# Hybrid Pressure Sensor Based on Carbon Nano-Onions and Hierarchical Microstructures with Synergistic Enhancement Mechanism for Multi-Parameter Sleep Monitoring

**DOI:** 10.3390/nano13192692

**Published:** 2023-10-01

**Authors:** Jie Zou, Yina Qiao, Juanhong Zhao, Zhigang Duan, Junbin Yu, Yu Jing, Jian He, Le Zhang, Xiujian Chou, Jiliang Mu

**Affiliations:** 1Science and Technology on Electronic Test and Measurement Laboratory, North University of China, Taiyuan 030051, China; 15735659591@163.com (J.Z.); zhaojuanhong2022@163.com (J.Z.); duanzhigang2050@163.com (Z.D.); yujunbin@nuc.edu.cn (J.Y.); jingyu5531@163.com (Y.J.); drhejian@nuc.edu.cn (J.H.); zhangle@nuc.edu.cn (L.Z.); chouxiujian@nuc.edu.cn (X.C.); 2School of Environment and Safety Engineering, North University of China, Taiyuan 030051, China; qiaoyina@nuc.edu.cn

**Keywords:** hybrid pressure sensor, CNOs@Ecoflex, hierarchical composite microstructure, synergistic enhancement mechanism, multi-parameter sleep monitoring

## Abstract

With the existing pressure sensors, it is difficult to achieve the unification of wide pressure response range and high sensitivity. Furthermore, the preparation of pressure sensors with excellent performance for sleep health monitoring has become a research difficulty. In this paper, based on material and microstructure synergistic enhancement mechanism, a hybrid pressure sensor (HPS) integrating triboelectric pressure sensor (TPS) and piezoelectric pressure sensor (PPS) is proposed. For the TPS, a simple, low-cost, and structurally controllable microstructure preparation method is proposed in order to investigate the effect of carbon nano-onions (CNOs) and hierarchical composite microstructures on the electrical properties of CNOs@Ecoflex. The PPS is used to broaden the pressure response range and reduce the pressure detection limit of HPS. It has been experimentally demonstrated that the HPS has a high sensitivity of 2.46 V/10^4^ Pa (50–600 kPa) and a wide response range of up to 1200 kPa. Moreover, the HPS has a low detection limit (10 kPa), a high stability (over 100,000 cycles), and a fast response time. The sleep monitoring system constructed based on HPS shows remarkable performance in breathing state recognition and sleeping posture supervisory control, which will exhibit enormous potential in areas such as sleep health monitoring and potential disease prediction.

## 1. Introduction

Sleep is an extremely vital physiological function that sustains human lives, among which adequate sleep plays a more essential role in enhancing learning ability, promoting physical and mental health, preventing chronic diseases, and most importantly, improving the quality of life [1,2]. However, with the accelerating pace of life and changing lifestyles, staying up late and insomnia [3] have become a norm, especially for particular groups of people. For the purpose of providing a sound and appropriate living environment, sleep monitoring is crucial. At present, the most mainstream means of sleep monitoring is polysomnography [4], but its shortcomings like high cost, time-consuming nature, and complicated process limit its further promotion. In recent years, researchers have devoted themselves to studying pressure sensors based on different mechanisms, such as magnetic [5,6,7], triboelectric [8,9,10,11], piezoelectric [12,13,14], piezoresistive [15,16], or capacitive [17,18], which have demonstrated tremendous potential in the field of sleep monitoring [8,12,19,20,21,22,23]. T. Uchiyama et al. have developed sensitive micromagnetic sensors based on magneto-impedance effect in amorphous wires, and they can monitor sleep through magnetoencephalography [7]. Lin Zhiming et al. reported a pressure-sensitive, large-scale, and washable smart textile based on triboelectric nanogenerator (TENG) array as bedsheet for real-time and self-powered sleep behavior monitoring [8]. Peng Min et al. designed a sleep biosignals detection system based on low-cost piezoelectric ceramic sensors. Under the action of the dynamic smoothing algorithm, piezoelectric ceramic sensors can be used for respiratory rate and heart rate detection with high accuracy [12].

An ideal pressure sensor has performance indicators with high sensitivity and a wide pressure response range. However, the high sensitivity of existing pressure sensors can only be maintained over a limited pressure response range. Lou et al. report the fabrication of a self-assembled 3D films platform for the first time [24]. The modular assembly of the rGO-encapsulated P(VDF-TrFe) nanofibers led to the fabrication of a highly sensitive pressure sensor (15.6kPa^−1^), but its pressure response range was only 1.2 Pa–55 kPa. Yang et al. prepared a pressure sensor based on an air-capsule TENG consisting of activated carbon/polyurethane and microsphere array electrodes, which has a pressure response range up to 7.27 MPa, but it is not outstanding in terms of sensitivity [25]. Hence, it is an enormous challenge to achieve the unification of high sensitivity and a wide pressure response range.

Preparation of microstructured functional layers is a critical factor in solving the above problems, and functional layer modification and microstructure preparation together determine the performance of pressure sensors [26,27]. The addition of functional fillers to the polymer matrix, which currently consist of nanoparticle fillers [28,29], liquid metals [30], and ionic liquids [31], is considered to be the most significant approach to functional layer modification. And the exploration of the type and content of functional fillers in the polymer matrix is essential for the electrical output of the functional layer. In addition, the fabrication methods of microstructures mainly include photolithographic template method [32,33], natural template method [34,35], 3D printing method [36,37,38], and so on. Despite the great progress in the research of microstructure preparation [34,39,40], there are still deficiencies in process complexity, manufacturing cost, and microstructure controllability. In view of this, the development of a simple, convenient, low-cost, and structurally controllable microstructure fabrication process is also imminent.

In this study, a strategy based on a synergistic material and microstructure enhancement mechanism is proposed for realizing the unification of high sensitivity with a wide pressure response range in a hybrid pressure sensor (HPS). Triboelectric pressure sensors (TPS) play an important role, and a butterfly mechanism designed using bionic strategies lays the foundation for the operation of TPS. The Ecoflex matrix was modified by adding carbon nano-onions (CNOs), and a simple, convenient, low-cost, and structurally controllable fabrication process was proposed for the preparation of CNOs@Ecoflex with hierarchical composite microstructures. In addition, the effect of material and microstructure synergies on the electrical properties of CNOs@Ecoflex was also investigated. The piezoelectric pressure sensor (PPS) is used to sense the weak external pressure and reduce the detection limit of the HPS. Finally, the TPS and the PPS are integrated into a structurally stable all-in-one device. The experiment shows that the HPS not only realizes the unity of high sensitivity and wide pressure response range, but also has fast response time, low detection limit, and high stability. The sleep monitoring system constructed based on HPS can be used for the acquisition of various health parameters such as respiratory status and sleeping posture, which provide scientific data support for medical diagnosis and will play an important role in the field of sleep health monitoring.

## 2. Experimental Section

### 2.1. Fabrication of the CNOs@Ecoflex

Preparation of hierarchical composite microstructured CNOs@Ecoflex films: In the first step, a mold with a prismatic structure on the surface is prepared using 3D printing. In the second step, a layer of conductive fabric is attached to the prismatic structure of the mold. In the third step, Ecoflex component A and CNOs are added to the beaker in a certain proportion and stirred with an electric mixer for about 3–5 h. Thereafter, Ecoflex component B is added and continued to be stirred for a period of time to form a mixed solution. In the fourth step, the mixed solution is poured into the pre-prepared molds and peeled off after one day of standing in a room temperature environment, at which time the hierarchical composite microstructured CNOs@Ecoflex film is prepared. In the fifth step, completed CNOs@Ecoflex is cut into a circular shape with a diameter of 1.2 cm as the negative electrode material for TENG. The detailed preparation procedure for hierarchical composite microstructured CNOs@Ecoflex is shown in the Appendix A (Appendix A).

### 2.2. Fabrication of the HPS

HPS consists of a press mechanism, a circular connector, a butterfly mechanism, a conductive fiber, a CNOs@Ecoflex film, a BTO@Ecoflex film, and a baseplate mechanism. The press mechanism, the circular connector, and the baseplate mechanism are made using 3D printing technology. Firstly, the conductive fiber is pasted on the bottom of the press mechanism as the friction positive electrode layer of TENG, and the BTO@Ecoflex is mounted on the bottom of the four support beams of the butterfly mechanism, and then the butterfly mechanism is placed in the groove on the bottom of the press mechanism, which is used for fixing the butterfly mechanism so that it can only move up and down. Secondly, a layer of double-sided conductive fibers is pasted on the upper side of the circular protrusion in the bottom plate mechanism, and CNOs@Ecoflex is pasted on the other side of the double-sided conductive fiber as the friction negative electrode layer of TENG. Finally, the press mechanism and the baseplate mechanism are encapsulated by circular connectors to make a closed body. Detailed preparation of BTO@Ecoflex is shown in the Appendix A (Appendix A).

### 2.3. Characterization and Measurement

The surface morphology of CNOs@Ecoflex and CNOs was characterized with scanning electron microscope (SEM, TESCAN MIRA LMS, TESCAN, Brno, Czech Republic). The crystal structures of Ecoflex and CNOs@Ecoflex were tested using X-ray Diffractometer (XRD, X*’*Pert PRO MPD). The constituent elements and chemical bonds of CNOs@Ecoflex were analyzed with X-ray Photoelectron Spectroscopy (XPS, Thermo Scientific K-Alpha, Thermo Fisher Scientific, Waltham, MA, USA). The molecular structure and chemical bonding composition of Ecoflex and CNOs@Ecoflex were tested with a Fourier Transform Infrared Spectrometer (FTIR, Thermo Scientific Nicolet iS20, Thermo Fisher Scientific, Waltham, MA, USA). The types and relative contents of elements in CNOs@Ecoflex were characterized with an X-ray Energy Dispersive Spectrometer (EDS). Output voltages were measured using a digital oscilloscope (Tektronix MSO2024B, Tektronix, Beaverton, OR, USA), and output currents were measured using a noise voltage preamplifier (Keithley 6514 system electrometer, Keithley, Cleveland, OH, USA). The potential distribution of the TENG was simulated using COMSOL 5.6, and the forces on the butterfly mechanism during operation were simulated using ANSYS 2023.

### 2.4. Statistical Analysis

Voltage and current data are preprocessed to reject outliers. Mean square error is used to detect outliers, and if the voltage or current data at a point exceeds three times the standard deviation, those points are defined as outliers. If there are missing values or outliers in a set of data, the set is discarded.

## 3. Results and Discussion

### 3.1. Structure and Fabrication of the HPS

Figure 1a illustrates the structure of HPS, which consists of a TPS and a PPS. TPS is based on the coupling effect of friction electricity and electrostatic induction, which consists of two kinds of friction materials with opposite polarity, and it largely determines the sensing performance of HPS. PPS is based on the pressure-sensitive effect, which can realize the conversion between mechanical energy and electrical energy. It is used as a complement to HPS to sense the external weak signals.

The positive and negative materials of the friction layer in TENG consist of conductive fabrics and CNOs@Ecoflex, respectively. In order to characterize the HPS with both wide pressure response range and high sensitivity, the friction layer was improved as follows. On the one hand, CNOs were chosen as the functional filler material, and their layered structures provide more effective interfaces and increase the efficiency of charge separation. This design enables them to withstand long time, high frequency friction, high durability, and stability. On the other hand, a simple, convenient, low-cost, and structurally controllable microstructure preparation method was proposed using 3D printing technology in combination with the natural template method, as shown in Figure 1b. A groove structure with a prismatic shape was fabricated using 3D printing technology, and subsequently conductive fiber with a fibrous structure was pasted into the grooves, and finally, a hierarchical composite microstructure mold with a prismatic-fibrous shape was prepared. Figure 1c shows a photograph of CNOs@Ecoflex, and the preparation of CNOs@Ecoflex is described in the Appendix A (Appendix A) and experiments section.

Based on the bionic strategy, the butterfly mechanism with a spring-like structure is designed. The butterfly mechanism is injection molded with flexible plastic, which has better elastic deformation and fatigue life, providing an important guarantee of the contact-separation TENG operation. When HPS is subjected to externally pressure, the pressing mechanism can only move vertically downward under the restriction of the circular connector. At this time, the butterfly mechanism deforms under the action of the press mechanism, and when the external force is released, the butterfly mechanism returns to its original shape. Figure 1d illustrates a photograph of the butterfly mechanism. Figure 1e shows a photograph of HPS, which is a fully encapsulated all-in-one structure with a high degree of structural stability that helps to adapt to a variety of operating environments. The structures and photographs of the various structural parts of HPS in the Appendix A (Appendix A).

Figure 2a represents the SEM image of CNOs, and it can be seen that CNOs present a multilayer structure with high specific surface area and large pore size distribution. Figure 2b,c represent the surface and cross-section microstructures of CNOs@Ecoflex, respectively. It can be seen from the surface image that the surface of CNOs@Ecoflex has a fiber-like structure, and from the cross-section image that CNOs@Ecoflex has a prismatic structure. The composition of CNOs@Ecoflex is analyzed with EDS, and Figure 2d shows that the major elements of CNOs@Ecoflex are C, O, and Si. In Figure 2e–g, CNOs are evenly distributed throughout the Ecoflex matrix. The XPS image (Figure 2h) is able to further confirm the presence of only C, O, and Si elements in CNOs@Ecoflex. As shown in Figure 2i, CNOs@Ecoflex showed obvious D and G peaks near 1350 cm^−1^ and 1580 cm^−1^. The D peak indicates the vibration caused by defects or impurities present in the CNOs, and the G peak indicates the vibration of sp^2^-hybridized carbon atoms in the CNOs. As shown in Figure 2j, there was no obvious characteristic peak of CNOs at 2*θ* = 25.56° for CNOs@Ecoflex, indicating that CNOs did not form large CNOs agglomerates in the Ecoflex matrix. In Figure 2k, the FTIR spectra of Ecoflex and CNOs@Ecoflex do not exhibit large shifts, and CNOs@Ecoflex does not produce strong new peaks, suggesting that there is no formation of new chemical bonds between Ecoflex and CNOs@Ecoflex, partly because of the low doping content of CNOs, and partly because the content of CNOs has no effect on the chemical structure of CNOs@Ecoflex.

### 3.2. Working Principle and Performance Characterization of the HPS

Figure 3a depicts the working principle of HPS, which contains two parts: the force analysis of the butterfly mechanism (Video S1) and the friction layer charge transfer mechanism. The operating mode of TENG is the contact-separation mode, and the butterfly mechanism is the basis for ensuring the successful operation of the contact-separation mode. When external pressure is applied to the HPS (Stage I), the butterfly mechanism deforms, and the surfaces of the friction layers come into contact with each other. Due to the triboelectric effect, a charge transfer occurs between the two materials at the contact area. The electrons on the surface of the conductive fiber are transferred to the surface of the CNOs@Ecoflex, so that the surface of the conductive fiber is positively charged and the surface of the CNOs@Ecoflex is negatively charged. When the external pressure on the HPS is gradually released (Stage II), the two friction layers of the HPS are in the separation stage due to the ability of the disc mechanism to automatically recover deformation, and a potential difference is formed between the two electrodes. When the external pressure on the HPS is completely released (stage III), the two friction layers of the HPS are separated to the maximum distance, and the open-circuit voltage reaches the saturation value. When a downward pressure is again applied to the HPS from the outside (Stage IV), the distance between the two friction layers of the HPS becomes smaller again, the potential difference between the two electrodes begins to decrease gradually, and the open-circuit voltage begins to decrease gradually. When the two friction layers of HPS are in complete contact with the external pressure, the open-circuit voltage decreases to zero. Figure 3b shows the simulation of the electric potential distribution when the friction layers are at different distances using COMSOL Multiphysics software.

Figure 4a shows the structure of the butterfly mechanism, and the angle between the “wings” of the butterfly mechanism and the vertical direction is θ. θ has an important influence on the mechanical properties of the butterfly mechanism. The relationship between θ and the maximum pressure on the butterfly mechanism, the maximum deformation of the butterfly mechanism, and the fatigue life of the butterfly mechanism is mainly investigated, and the results are shown in Figure 4b. The smaller the θ, the lower the maximum pressure on the butterfly mechanism placed under during operation, which means it will have a longer working life. However, the contact-separation distance of the TENG friction layer becomes smaller, which seriously affects its electrical output performance. Compression cycle experiments were carried out on the butterfly mechanism with different θ, and the number of compressions was 100,000. The results showed that the deformation of the butterfly mechanism with 30°~45° was 0.3%, which means that it has a significantly excellent fatigue life. In summary, the θ of the butterfly mechanism is chosen to be 45°. This ensures the electrical output performance of the HPS and provides itself with a good fatigue life. A detailed experimental description of the mechanical properties of the butterfly mechanism can be found in the Appendix A (Appendix A).

In order to evaluate the effect of different contents of CNOs on the performance of CNOs@Ecoflex, an electrical experimental platform was built to test the electrical properties of CNOs@Ecoflex. Figure 4c shows the dielectric constant and dielectric loss of CNOs@Ecoflex. Compared with Ecoflex, the dielectric constant of the CNOs@Ecoflex increased from 1.68 to 18.06. This phenomenon is attributed to the high electrical conductivity and surface area of the CNOs, which can improve the conductive properties and charge transfer capability of Ecoflex, ultimately increasing the dielectric constant of CNOs@Ecoflex. As shown in Figure 4d, the $V_{OC}$ (≈98.73 V) and $I_{SC}$ (≈4.42 µA) of CNOs@Ecoflex with 5% CNOs content are enhanced by a factor of 1.83 and 2.04, respectively, compared with Ecoflex. The electrical enhancement effect can be obtained using the following equation [41,42]:(1)$$\sigma =\frac{V\varepsilon_{ri}\varepsilon_{0}}{d_{i}}$$
where σ is the surface charge density, V is the potential difference between the two electrodes, $\varepsilon_{0}$ is the vacuum dielectric constant, and $d_{i}$ and $\varepsilon_{ri}$ are the thickness and dielectric constant of the dielectric material between the electrodes, respectively. From this equation, the maximum transferred charge density increases with the increase in dielectric constant, and the increase in surface charge density ultimately enhances the electrical output of CNOs@Ecoflex. However, CNOs are rigid nanofillers, and too high a proportion of CNOs filling in Ecoflex tends to cause the phenomenon of regional aggregation [43,44], which increases the interaction force between particles. The regional aggregation will reduce the mechanical and electrical properties of CNOs@Ecoflex, thereby weakening the interfacial bonding [45]. Therefore, the electrical performance of CNOs@Ecoflex are reduced after the CNOs content exceeds 5%.

The preparation of different microstructures has proven to be an effective means of improving the sensing performance of devices [46]. As shown in Figure 4e, the electrical output of four different microstructures of CNOs@Ecoflex is compared. The SEM image provides insight into the four different microstructures, with the hierarchical composite microstructure being a combination of two single microstructures. It can be seen that the CNOs@Ecoflex with hierarchical composite microstructures prepared in this paper has the best electrical output performance compared to CNOs@Ecoflex with no microstructure and with a single microstructure, and the $V_{OC}$ and $I_{SC}$ are enhanced by factors of 1.52 and 1.29, respectively, and the enhancement of the electrical performance of the CNOs@Ecoflex provides a significant improvement in the sensing performance of the device.

Pressure response range refers to the range of pressure intervals over which a pressure sensor can operate stably. Depending on the response range of the device, the sensor can be used in various applications. Sensitivity is one of the most important performance parameters of pressure sensors and is also closely related to the pressure response range. As shown in Figure 5a–d, the pressure response range of HPS is 10–1200 kPa (1–120 N), and the minimum pressure detection limit is 10 kPa (1 N). The voltage sensitivity of the HPS was 0.566 V/10^4^ Pa (0.566 V/N) at 10 to 50 kPa (region 1), 2.46 V/10^4^ Pa (2.46 V/N) at 50 to 600 kPa (region 2), and 0.61 V/10^4^ Pa (0.61 V/N) at 600 to 1200 kPa (region 3). HPS has a voltage linearity of 0.997 over the range of 10 to 50 kPa (region 1); 0.992 over the range of 50 to 600 kPa (region 2); and 0.981 over the range of 600 to 1200 kPa (region 3).

Figure 5e shows the output voltage of HPS at a working frequency of 1–3 Hz (the frequency and force are constant), and the response times of HPS at frequency of 1–3 Hz are 107, 66, 48, 39, and 32 ms, which are able to adapt to the environment of different frequencies. Stability is the change in sensing performance of a pressure sensor under conditions of extended operation. It is an important parameter for evaluating whether the device can be used in practical applications for a long time and multiple cycles. As shown in Figure 5f, durability experiments have been conducted on HPS, and the $V_{OC}$ and the $I_{SC}$ are intercepted after 100,000 times of continuous working. The results show that HPS does not change significantly after a long period of time, which proves that HPS has high durability and stability. Therefore, the high intensity and long-term working conditions do not lead to the degradation of the sensing performance of HPS.

In conclusion, HPS designed in this paper is a pressure sensor with excellent overall performance, which may be attributed to the following factors: (1) the synergistic effect of TPS and PPS, (2) CNOs and hierarchical composite microstructures synergistic enhancement mechanism, and (3) the butterfly mechanism with good elastic deformation and fatigue life constructed based on bionic strategies. Table 1 shows the performance comparison between HPS and previous studies. As shown in Table 1, HPS has a wide pressure measurement range compared to Refs. [21,22,43,45]. HPS has high sensitivity compared to Refs. [22,43,44,45]. HPS has a low detection limit compared to Refs. [44,45].

### 3.3. Application of HPS in the Field of Sleep Monitoring

With our lifestyle change, the average amount of sleep people get is decreasing year by year, and the quality of sleep is getting worse. As shown in Figure 6a, deterioration in sleep quality will cause great harm to the physical and mental health of human beings, and there is an urgent need for sleep monitoring in this group. However, traditional monitoring means have the disadvantages of high price, cumbersome process, and poor portability, which seriously limit the popularity of sleep monitoring means. Based on the above analysis, a low-cost, easy-to-operate, and portable multi-parameter sleep monitoring system is designed in Figure 6b, where the array-dispersed HPS can not only sense the pre-breathing situation under different sleep conditions, but also record the sleeping posture during sleep in real time. The collected sleep parameter information can be processed and displayed through a visual interface (Figure 6c) to scientifically analyze the human sleep condition and predict possible human health problems. In addition, the biocompatibility of HPS is also discussed, and detailed information is provided in the Appendix A (Appendix A).

During sleep breathing, the human shoulder usually experiences some degree of movement. As shown in Figure 7a, during inspiration, the volume of the thoracic cavity increases due to thoracic expansion and diaphragmatic contraction, at which point there is a rising action of the shoulders. During exhalation, the volume of the thoracic cavity decreases, and a descending movement of the shoulders occurs. Based on the above phenomena, the pressure situation of the shoulders is measured to reflect the breathing situation in different sleep conditions. Figure 7b shows the breathing curves under four different sleep conditions, where shallow breathing has the fastest respiratory rate and deep breathing has the largest respiratory amplitude. The collected curves were recognized by an improved VGG network model, and the schematic diagram of the improved VGG network model is shown in Figure 7c. Figure 7d shows the accuracy and loss values of the improved VGG network model during training, and the accuracy of the improved VGG network model for system identification was stable at 100% after the third round of training. Finally, the prediction samples are input to the improved VGG network model, and the prediction results are shown in Figure 7e. The improved VGG network model can effectively discriminate between the four different sleep breathing behaviors with 100% accuracy (Video S2). Detailed information on the prediction process is provided in the Appendix A (Appendix A).

Sleep posture is an important parameter for human sleep quality assessment, and sleep posture monitoring can not only help people better understand their sleep habits and sleep quality, but also improve their sleep experience and health by adjusting their sleep posture. As shown in Figure 8a, there are five common sleeping postures including supine, left recumbent, right recumbent, prone, and curl up. When the human body is in different sleeping postures, there are obvious differences in the pressure of the head and shoulders on the pillow and mattress. As shown in Figure 8b, the pressure values of the arrayed HPSs were varied in different sleeping postures, and different sleeping postures could be accurately distinguished by the pressure values of each HPS (Video S3). In addition, the volunteers were monitored for up to 3 h of sleeping posture, and the results are shown in Figure 8c. The volunteers rolled over six times during the sleep period, with the supine being the longest, accounting for 63% of the total sleep time. In addition, a detailed comparison of the sleep monitoring system designed in this work with existing sleep monitoring systems was made, and the results of this comparison are provided in the Appendix A (Appendix A).

## 4. Conclusions

In conclusion, this work proposes a hybrid pressure sensor with CNOs and hierarchical composite microstructure synergistic enhancement mechanism for multi-parametric human health sleep monitoring. It integrates the TPS and the PPS. On the one hand, for the TPS, the butterfly mechanism constructed based on the bionics strategy has strong elastic deformation and fatigue life. A simple, low-cost, and structurally controllable microstructure preparation method is also proposed. Under the synergistic effect of CNOs and hierarchical composite microstructures, the prepared CNOs@Ecoflex film exhibits excellent electrical properties, and its $V_{OC}$ and $I_{SC}$ are increased by 2.796 and 2.631 times, respectively. On the other hand, the PPS, as a complement to the TPS, not only broadens the pressure corresponding range, but also reduces the pressure detection limit. Experiments have shown that HPS has a high sensitivity of 2.46 V/10^4^ Pa (50–600 kPa) and a wide detection range of up to 1200 kPa, with a minimum detection limit of 10 kPa. In addition, HPS has excellent cyclic stability (more than 100,000 cycles) and can maintain excellent electrical performance, stable response frequency, and fast response time under different external forces and frequencies. The sleep monitoring system based on HPS shows excellent performance in breathing monitoring and sleeping posture monitoring, and the improved VGG network model can accurately identify breathing under different sleeping conditions with an accuracy of 100%. The array deployment of HPS can accurately reflect the sleeping posture of the human body and can record changes in sleeping position in real time. HPS shows great potential in sleep health monitoring, potential physiological disease prediction, and other human–computer interaction fields.

## Figures and Tables

**Figure 1 nanomaterials-13-02692-f001:**
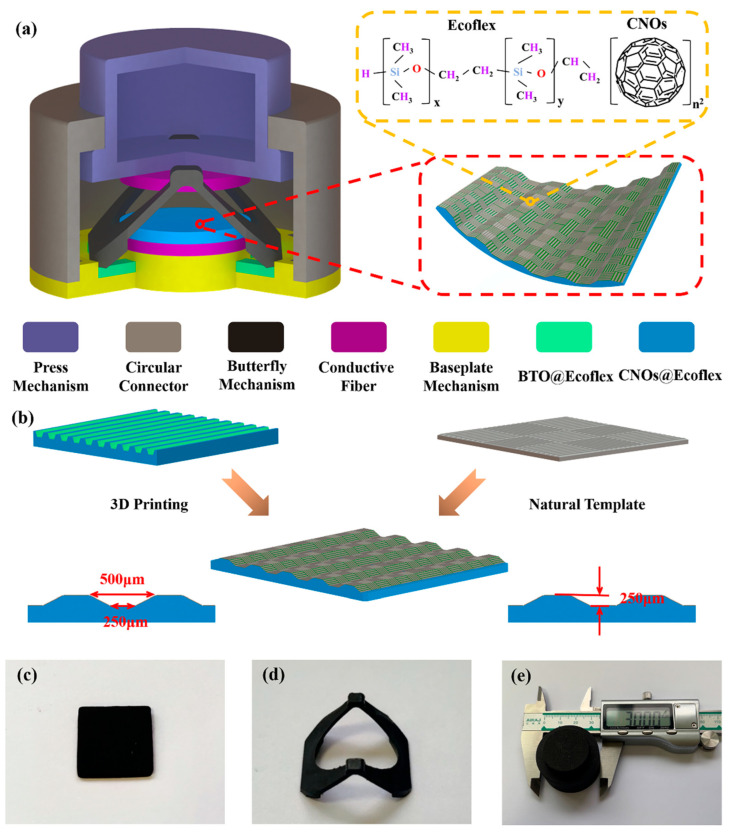
Structure of HPS. (**a**) Schematic of HPS. (**b**) Hierarchical composite microstructure preparation. (**c**) Photograph of the CNOs@Ecoflex. (**d**) Photograph of the butterfly mechanism. (**e**) Photograph of HPS.

**Figure 2 nanomaterials-13-02692-f002:**
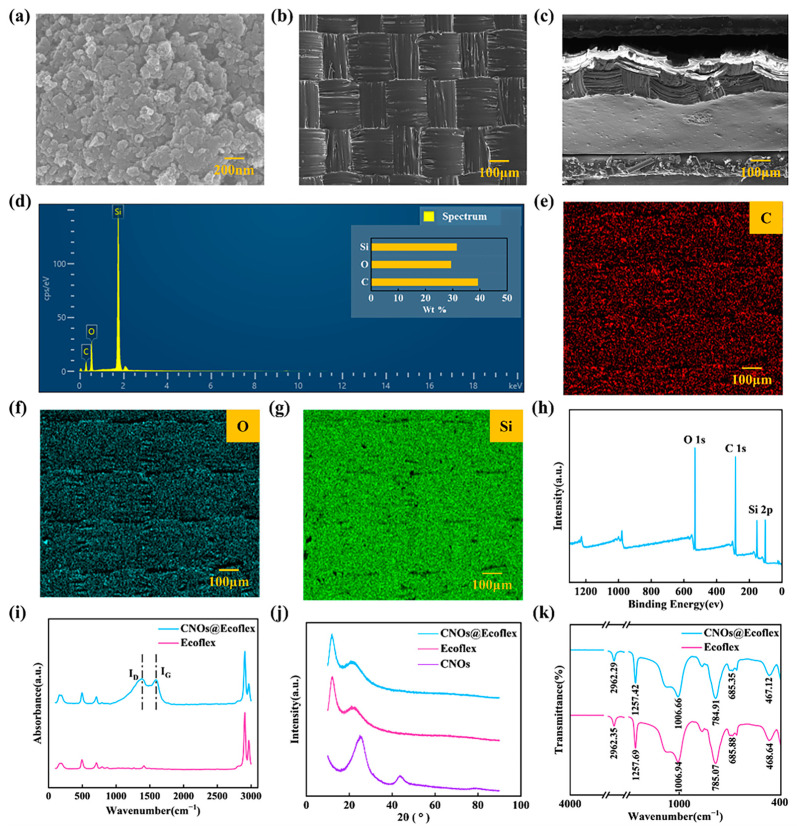
Characterization of CNOs@Ecoflex. (**a**) SEM image of the CNOs. (**b**,**c**) SEM image of the CNOs@Ecoflex. (**d**–**g**) EDS spectra of CNOs@Ecoflex. (**h**) XPS spectra of CNOs@Ecoflex. (**i**) Raman diagram of CNOs@Ecoflex and Ecoflex. (**j**) XRD patterns of CNOs@Ecoflex and CNOs. (**k**) FTIR spectra of CNOs@Ecoflex and Ecoflex.

**Figure 3 nanomaterials-13-02692-f003:**
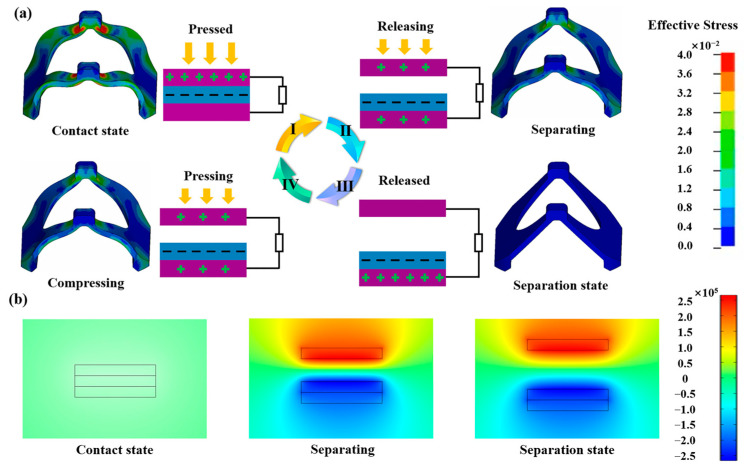
Working principle of the HPS. (**a**) Force analysis of butterfly mechanism and friction layer charge transfer mechanism. (**b**) Simulation of potential distribution in friction layer.

**Figure 4 nanomaterials-13-02692-f004:**
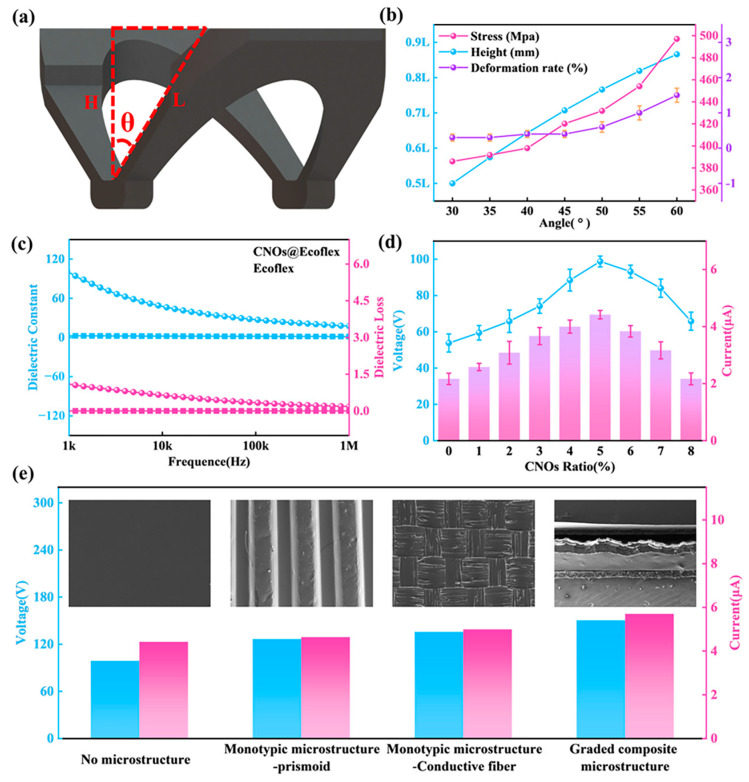
Structural improvement and material modification of HPS. (**a**) Schematic structure of the butterfly mechanism. (**b**) Mechanical properties of the butterfly mechanism. (**c**) Dielectric constant, dielectric loss of Ecoflex and CNOs@Ecoflex. (**d**) Electrical output properties of CNOs@Ecoflex. (**e**) Electrical output of CNOs@Ecoflex with different microstructures.

**Figure 5 nanomaterials-13-02692-f005:**
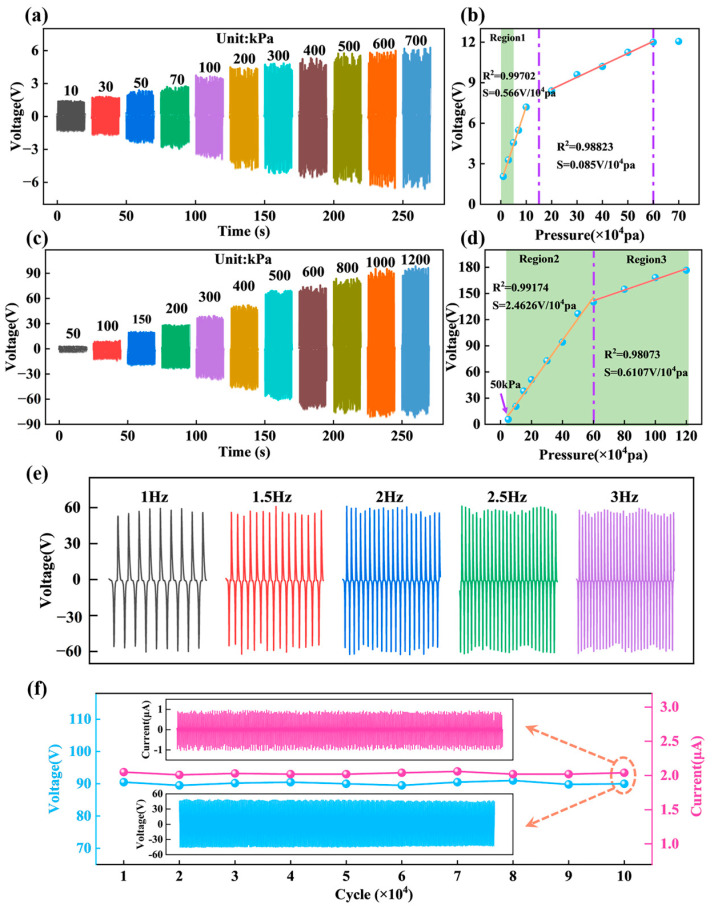
Sensing performance of HPS. (**a**) Pressure response range of TPS. (**b**) Sensitivity of TPS. (**c**) Pressure response range of the PPS. (**d**) Sensitivity of PPS. (**e**) Electrical output of the HPS at 1–3 Hz working frequency. (**f**) Durability experiment of HPS.

**Figure 6 nanomaterials-13-02692-f006:**
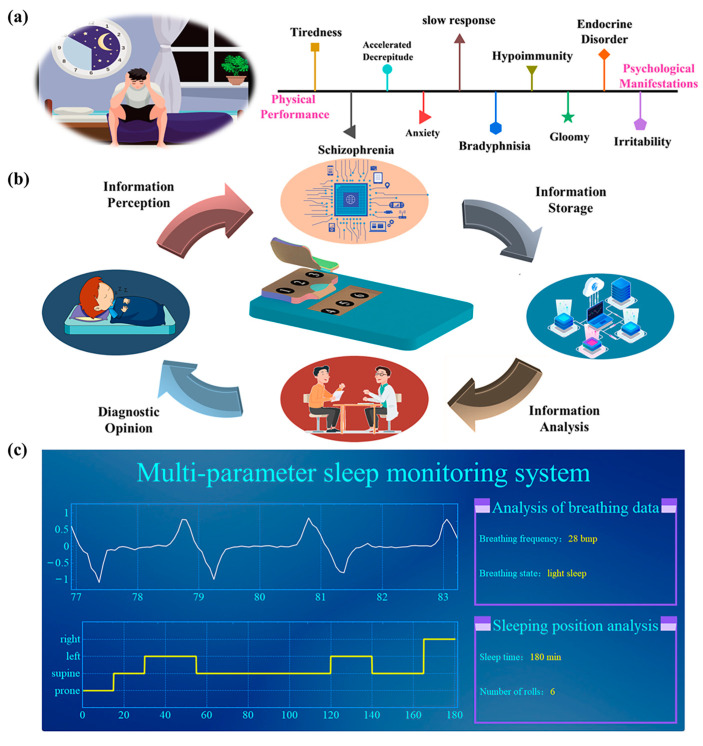
Sleep monitoring system. (**a**) Physical and psychological hazards of poor sleep quality. (**b**) Schematic diagram of sleep monitoring system based on array HPS. (**c**) Visualization interface of the sleep monitoring system.

**Figure 7 nanomaterials-13-02692-f007:**
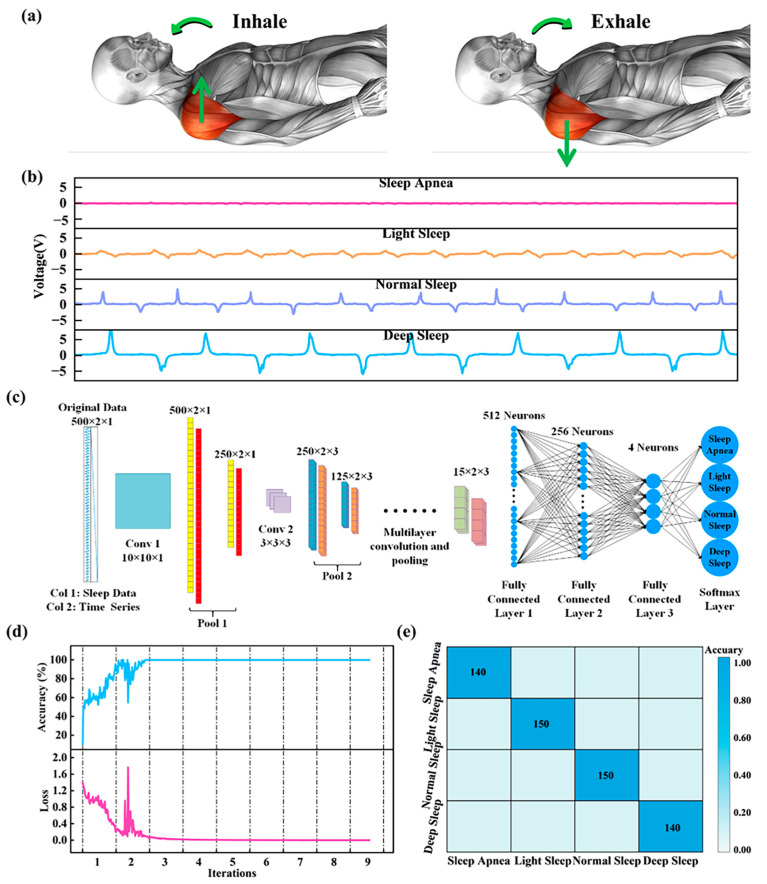
Breathing monitoring and prediction in different sleep states. (**a**) Human shoulder motion model during sleep breathing (**b**) Breathing curves for four different sleep states. (**c**) Schematic of the improved VGG network model. (**d**) Accuracy and loss values of the improved VGG network model in training. (**e**) Prediction results of the improved VGG network model.

**Figure 8 nanomaterials-13-02692-f008:**
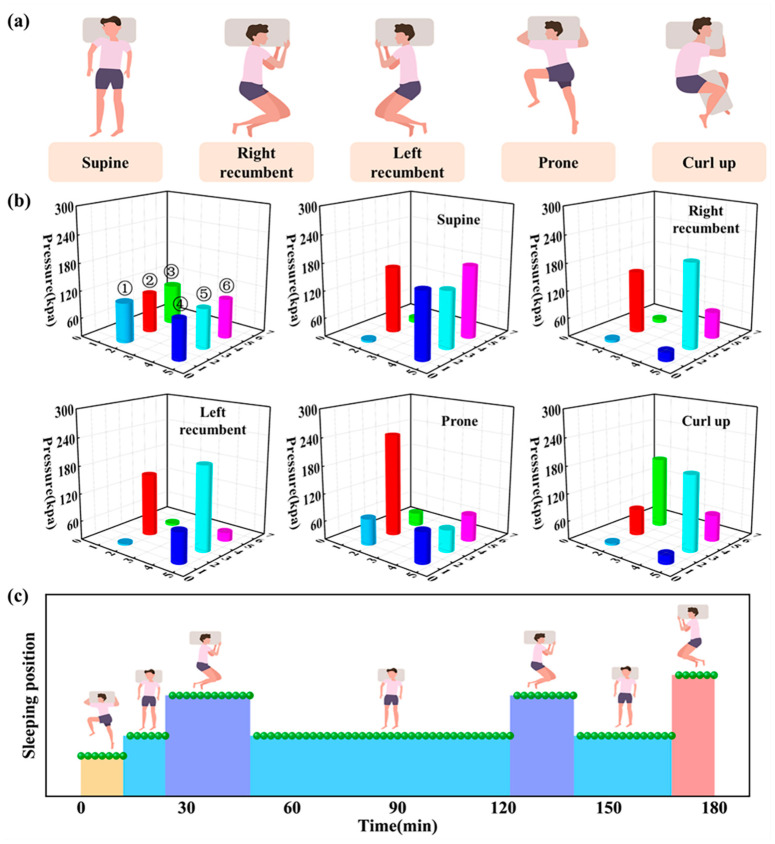
A sleeping posture monitoring system. (**a**) Five types of common sleeping postures. (**b**) Pressure values of HPSs in different sleeping postures. (**c**) Sleeping postures of the volunteers during 3 h.

**Table 1 nanomaterials-13-02692-t001:** Comprehensive comparison of pressure sensors based on different mechanisms.

Sensor Mechanism	Sensitive Material	Pressure Response Range	Limit of Detection	Sensitivity	References
Triboelectric Sensors	P(VDF-TrFe) and rGO	1.2 Pa–55 kPa	1.2 Pa	15.6 kPa^−1^	[24]
Triboelectric Sensors	Arbon and polyurethane	Up to 7.27 MPa	-	0.2 V/N	[25]
Triboelectric Sensors	Elastomer and ionic hydrogel	1.3–101.2 kPa	1.3 kPa	0.013 kPa^−1^	[47]
Piezoelectric Sensor	PVDF and BTO	12–243 N	12 N	0.775 V/N	[48]
Piezoelectric Sensor	BTO	5–50 N	5 N	0.05 V/N	[49]
Hybrid Sensor	CNOs and Ecoflex	10–1200 kPa (1–120 N)	10 kPa (1 N)	0.246 kPa^−1^/ 2.46 V/N	This work

## Data Availability

Data are contained within the article and the Appendix A.

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
