# Peer review of "Hybrid Pressure Sensor Based on Carbon Nano-Onions and Hierarchical Microstructures with Synergistic Enhancement Mechanism for Multi-Parameter Sleep Monitoring"

_nanomaterials, 2023, doi:10.3390/nano13192692_

Round 1

Reviewer 1 Report

Submitted manuscript is devoted to the development of particular type of sensor platform for advanced control of sleeping state. The subject is interesting and was attracting special attention for the last 2 decades. With this respect the poor description of the topic in the Introduction causes quite a strange impression. Pressure can be measured by many different devices and in very different scales – from very high to very low levels. For instance, at the level of microfluidic chambers or wearable devices the magnetic sensors are very efficient (Benavides et al. DOI 10.1088/1361-6501/aaa27f; Fernandez et al. DOI https://doi.org/10.1186/1556-276X-7-230; etc.). Group of Prof. Mohri had systematically investigated the possibility to use sensor platform based on giant magnetic impedance for advanced control of sleeping state and measurements of magnetocardiogram and magnetoencephalogram (Uchiyama https://10.1109/TMAG.2012.2198627, etc.). Authors should place their results in a way of  careful analysis of previous contributions.

Work is badly written because they use abbreviations like CONs even in the title. It very narrow known abbreviation. For example, in computer programming, CONs is a fundamental function in most dialects of the Lisp programming language used for constructing memory objects which hold two values. The goal of the study is not written in the clear way at the and of Introduction. Grammar requires many corrections and refinements. For instance, the statement “The volunteers rolled over six times during the sleep period, in which the su- 367 pine was the longest, accounting for 63.33% of the whole sleep time” gives the 0.03% significant numbers. What is biologically accepted time of the transition from sleep to wakefulness?

Figure 4 b (very important) has no error bars. Table 1 has wrong writings of significant numbers. Discussion must show comparison with existing sleep monitoring systems and provide careful comparison.

Biocompatibility and energetic efficiency of the proposed materials is not discussed.

Extensive editing of English language required.

Author Response

We would like to acknowledge you for having spent time on handling and reviewing this manuscript. We studied carefully your insightful comments, and a number of necessary changes were made and highlighted in red in the manuscript based on your comments. We believe that these changes can fully address all concerns of reviews. In the next, we provide point-by-point responses to your comments and concerns.

Reviewer 2 Report

The authors presented a large, complex work, including the manufacture of an effective pressure sensor, the design of a mechanism for monitoring breathing, and the development of a sleep monitoring system based on the presented mechanism. The article is well written, but there are some comments.

1. Is it possible to clarify how the mechanical properties of the mechanism were studied (For example, Fig. 4b). Also, I couldn’t find what material the butterfly was made from?

2. You tested the stability of the mechanism against 10,000 compression cycles and got good results. However, assuming on average that a person takes 10 breaths per minute, that's 600 per hour or 14,440 per day. So the value of 10,000 cycles does not seem to be representative of the long-term stability of the mechanism. Is it possible to provide any longer-term stability data for more cycles?

3. "...an electrical experimental platform was built to test the electrical properties of CONs@Ecoflex. Figure 4c shows the dielectric constant and dielectric loss of CONs@Ecoflex. Compared with Ecoflex, the dielectric constant of the CONs@Ecoflex has increased from 1.68 to 18.06."

(lines 245-248)

A few notes here.

3.1. Wouldn't it be better to write here that they would test dielectric properties rather than electrical ones?

3.2. In Fig. 4c, the values of the dielectric constant are completely different than in the text, in addition, they are negative. Check it.

3.3. What electrical contacts were used and how were they attached to the mechanism during electrical testing?

"In the third step, Ecoflex component A and CONs were added to the beaker in proportion and stirred with an electric mixer for about 3-5 h." (lines 88-89)

May be, "in a certain proportion"?

Author Response

(The authors gave the same response as above.)

Round 2

Reviewer 1 Report

Work can be published in this state.

Minor changes.

Reviewer 2 Report

Authors have made noted corrections; the article can be published as presented.